# Prediction of Tumor Development and Urine-Based Liquid Biopsy for Molecule-Targeted Therapy of Gliomas

**DOI:** 10.3390/genes14061201

**Published:** 2023-05-30

**Authors:** Michihiro Kurimoto, Yumi Rockenbach, Akira Kato, Atsushi Natsume

**Affiliations:** 1Department of Neurosurgery, Aichi Children’s Health and Medical Center, Obu 464-8710, Japan; 2Institute of Innovation for Future Society, Nagoya University, Nagoya 464-8601, Japan

**Keywords:** brain tumor, glioma, gene mutation, mathematical model, isocitrate dehydrogenase, microRNA

## Abstract

**Simple Summary:**

This review article presents novel therapies being developed for central nervous system cancers. Based on recent studies of our own and others, we have enumerated six molecules, which upon mutation are regarded as the primary cause of cancer development and are as follows: isocitrate dehydrogenase, telomerase reverse transcriptase, BRAF, O^6^-methylguanine-DNA methyltransferase, histone3-lysine27/histone3-guanine34, and NTRK/ROS1. Molecule-targeting drugs have been developed based on tumor initiation and how these mutations are involved in tumor development and progression.

**Abstract:**

The timing of the acquisition of tumor-specific gene mutations and the systems by which these gene mutations are acquired during tumorigenesis were clarified. Advances in our understanding of tumorigenesis are being made every day, and therapies targeting fundamental genetic alterations have great potential for cancer treatment. Moreover, our research team successfully estimated tumor progression using mathematical modeling and attempted early diagnosis of brain tumors. We developed a nanodevice that enables urinary genetic diagnosis in a simple and noninvasive manner. Mainly on the basis of our research and experience, this review article presents novel therapies being developed for central nervous system cancers and six molecules, which upon mutation cause tumorigenesis and tumor progression. Further understanding of the genetic characteristics of brain tumors will lead to the development of precise drugs and improve individual treatment outcomes.

## 1. Introduction

Although numerous multidisciplinary therapeutic strategies that combine surgical resection, chemotherapy, radiation therapy, and other treatment approaches have been attempted for central nervous system (CNS) cancers, several tumor types are still characterized by extremely poor prognosis. A typical example is a glioblastoma, which is classified by the World Health Organization (WHO) as a grade 4 tumor type and has a very poor prognosis, with a 5-year survival rate of approximately 10% [1,2]. Moreover, regardless of the tumor type, in several cases, treatment options are extremely limited owing to the anatomical location of the tumors. Furthermore, brainstem glioma, which is common in children and characterized by a high degree of pathological malignancy, has limited treatment options, with a 2-year survival rate of only 10% [3]. In these tumors with poor prognosis, it is imperative to develop novel therapeutic approaches, as well as improvements in surgical resection techniques such as supra-total resection and endoscopic tumor resection.

Several novel therapies for CNS cancers are currently under development. Several advances in genome analysis have been made and studies on how genetic aspects are related to the development, progression patterns, treatment responsiveness, and recurrence of cancers after treatment have been reported. Molecule-targeting drugs and several other novel therapies have also been developed as part of this process. In this review, we have presented new perspectives on the development and progression of CNS cancer and its associated molecular mechanisms.

## 2. Mathematical Model of Tumor Development

Diffuse glioma, which accounts for approximately 80% of CNS tumors, is classified as a grade 2–4 tumor type by the WHO [4]. Although grade 2 tumors are generally considered low-grade malignancies, they often have poor prognoses, owing to malignant transformations that occur over time during treatment or at recurrence [4]. Such malignant transformations have been attributed to therapeutic interventions, such as chemotherapy and radiotherapy, which are performed as standard treatments [5,6]. Grade-2 gliomas are responsive to radiotherapy plus chemotherapy with temozolomide (TMZ) or the regimen of procarbazine, lomustine (CCNU), and vincristine (known as PCV) [7,8,9,10]. However, the timing of these therapies after surgery is controversial. If the clinical course of an individual tumor is predicted based on the tumor’s genetic background, we would be able to estimate the best timing for surgery. The advent of genome-wide analysis technologies has revealed the characteristic genetic lesions in a variety of tumors including LGGs, and the results well imply clinical outcomes [11,12,13]. For example, mutations in the genes encoding isocitrate dehydrogenase (IDH) 1 and 2 (*IDH1* and *IDH2*) account for approximately 80% of grade-2 gliomas and represent one of the fundamental and earliest molecular events in the genesis of these tumors [14,15,16,17]. The IDH-mutant grade 2-gliomas are classified into two subtypes which demonstrate different clinicopathological and genetic characteristics depending on the presence of co-deletion of the short arm of chromosome 1 (1p) and the long arm of chromosome 19 (19q) (1p/19q) according to the revised WHO classification [4]. Longitudinal studies of IDHmut-gliomas have identified several genetic and epigenetic alterations more commonly in recurrent tumors with malignant transformation compared to primary tumors [18,19]. These genetic and epigenetic insights into tumors showed sequential events happening within single cells and the clinical behaviors of an individual. Several attempts have been made to use mathematical models of the routes taken by these cancers, from tumor development to malignant transformation, to the end of making inferences regarding the factors involved and their timing. Among a number of interdisciplinary studies, two mathematical models accounted for the malignant transformation of Grade-2 gliomas [20,21]. However, these studies analyzed a small number of patients who underwent surgery alone, and estimated growth parameters, but failing due to rough prediction of tumor progression. Conversely, our team used time-series data to construct a mathematical model that accounted for the effects of therapeutic interventions on factors such as tumor volume (calculated from images) and treatment history, using 276 cases of IDH-mutant glioma. Using comprehensive genetic analyses for each case, we were able to estimate the effects of therapeutic interventions on tumor progression and malignant transformation, considering the period from tumor onset to diagnosis and the timing of therapeutic interventions [22]. Consequently, the model revealed that prompt adjuvant chemoradiotherapy prolonged malignant transformation-free survival if the size of tumors was ≤50 cm^3^. Furthermore, optimal treatment differed according to genetic alterations for large tumors (>50 cm^3^); adjuvant therapies prolonged malignant transformation-free survival in IDH^mut^/1p19q^noncodel^ tumors. Importantly, phosphoinositide 3-kinase mutation would be a key accelerator in IDH^mut^/1p19q^codel^ tumors, which increased postoperative proliferation rate and shortened malignant transformation-free survival. This mathematical model is an important tool that can be used to trace tumor development.

## 3. Urine-Based Genetic–Molecular Diagnosis

Owing to the findings related to tumor development described, genetic analyses are increasingly being used in the field of tumor diagnosis, which relies on pathological diagnosis. This led to a significant reorganization of the WHO classification system in 2021 [23].

Liquid biopsy has offered an attractive alternative for diagnosing and monitoring cancer in the past decade. Since tissue biopsies can only collect a portion of cancer tissue, biased information due to spatial heterogeneity within and between tumors is inevitable, making it difficult to obtain a complete picture of the tumor tissue [24,25,26]. On the other hand, since liquid biopsy collects body fluids such as blood and urine that contain tumor-related information, it can obtain information on a wide range of tumor tissues without being affected by spatial heterogeneity [27]. The main advantage of liquid biopsy is that only body fluids such as blood or urine need to be collected, thus minimizing pain and physical burden. Due to the ease of specimen collection in liquid biopsy, long-term follow-up testing for early detection and prediction of cancer recurrence is possible, and even if a cancer acquires new drug resistance during treatment, testing of cancer response to drugs would be easily performed. The advent of liquid biopsy provides a noninvasive yet repeatable method to track cancer in real time, leading to improved cancer management and outcomes. Advances in measurement and analysis technologies such as quantitative polymerase chain reaction (qPCR), next-generation sequencing (NGS) and mass spectrometry have expanded the scope of application of liquid biopsy for cfDNA and EVs in body fluids. As for attempts to detect genetic mutations by analyzing cfDNA in blood, what began as an assay to detect KRAS/BRAF mutations associated with resistance to EGFR antibody therapy in colorectal cancer patients has evolved into the development of a commercial panel test for a large cohort of more than 10,000 patients [28,29]. Other gene panel tests that can detect more than 70 genetic mutations from a blood sample include Foundation Medicine’s FoundationOne Liquid CDx and Guardant Health’s Guardant360, which are in development with FDA breakthrough device designation [30]. Our group reported on the scalability of zinc oxide (ZnO) nanowire performance, in which we have found the capture of cell-free DNA (cfDNA) via hydrogen bonding in addition to the previously reported capture of extracellular vesicles via surface charge. The ZnO nanowire devices have previously been developed to collect urinary EVs [31,32,33] and profile urinary EV-derived microRNAs (miRNAs) for mass screening of CNS tumors [34]. cfDNA can be recovered from the urine of patients with brain tumors followed by NGS-mediated detection of cfDNA mutations, thus leading to identification of molecular subtypes of brain tumors.

## 4. Molecule-Targeted Therapy

Studies on the initiation of tumor-specific gene mutations and how these gene mutations occur during tumorigenesis are being carried out. Molecule-targeting drugs have been developed based on tumor initiation and how these mutations are involved in tumor development and progression.

While a number of genes such as *TP53*, *PTEN* and *CDKN2A/B* are critical to tumor malignancy, in this review we focus on six druggable molecules: isocitrate dehydrogenase (IDH), telomerase reverse transcriptase (TERT), BRAF, O^6^-methylguanine-DNA methyltransferase (MGMT), histone3-lysine27 (H3K27)/histone3-guanine34 (H3G34), and NTRK/ROS1. The mutation, deletion, or amplification of these genes promote cancer development, where the altered genes and dysregulated transcription exerts a gain of function.

### 4.1. Isocitrate Dehydrogenase

Although it was first reported by Vogelstein et al. as a WHO-grade 4 glioblastoma; recent analyses have shown that most WHO grade 2 and 3 astrocytoma/oligodendrogliomas are characterized by IDH1/2 mutations [35]. Specifically, wild-type IDH1/2 uses NADP+ as a coenzyme to produce α-ketoglutaric acid (αKG) from isocitric acid, whereas mutant IDH1/2 uses NADPH as a coenzyme to produce oncometabolite D-2-hydroxyglutaric acid (D2HG) from αKG. Furthermore, D-2-HG competitively inhibits αKG–dependent dioxygenases, such as HIF propyl4 hydroxylase, collagen propyl4 hydroxylase, and TET2 hydroxylase, and as a consequence stabilizes HIF-1/2α protein, and impairs collagen maturation, cytosine demethylation, and histone demethylation [36,37,38,39,40]. The accumulation of D-2-HG leads to early gliomagenesis, followed by clonal expansion through epigenetic dysregulation [41,42]. The mutant-IDH induced methylation, a factor known as glioma CpG island methylator phenotype (G-CIMP), could probably result from the effects of IDH mutation on epigenetic regulation, which contributes to tumor development [10,36,43] (Figure 1).

Furthermore, this mutation occurs during tumor formation in the earliest stages of tumor development; thus, according to WHO 2021 glioma classifications, the IDH mutation status should be examined when classifying glioblastoma and astrocytomas/oligodendrogliomas (Table 1).

Mutant IDH inhibitors have been used to treat cancer stem cells (CSCs) associated with myeloid leukemia [44,45]. Phase I and II trials involving the use of a selective mutant IDH1 inhibitor (DS-1001) for the treatment of recurrent IDH-mutant gliomas have been conducted [44]. Therefore, if therapeutic effectiveness is confirmed, the use of IDH inhibitors could become a treatment option for a wide range of tumor types. Hence, the results of these trials are eagerly awaited.

### 4.2. TERT

Telomerase reverse transcriptase (TERT) that prolongs telomere length is generally thought to maintain telomere length and thus genome stability but is rarely expressed in normal cells. Increased expression of TERT has been reported in a number of tumor types. Theoretically, the suppression of TERT gene expression leads to decline in telomerase activity during the differentiation of regular somatic cells. This is not, however, the case in tumor cells. Delayed TERT re-expression forcibly maintains telomeres and prevents cell death. Although several aspects of this mechanism remain unclear, a two-stage model in which cells are made “immortal” via the induction of selective telomere prolongation and genomic instability owing to shortened telomere fusion and the reactivation of telomerase activity has been proposed [45]. The majority of TERT gene mutations occur in the promoter region, especially, at 124 or 146 bases upstream from the start codon. The mutations paradoxically upregulate the expression of TERT.

In gliomas, TERT gene mutation is positively correlated with IDH mutation and 1p19q co-deletion [11], therefore it may be involved in early stage tumor development (Table 1). This mutation also plays an important role in glioma differentiation. Astrocytomas and oligodendrogliomas both develop by acquiring IDH mutations at an early stage, as described above, but further develop into astrocytomas by sequential mutations in TP53 and ATRX, whereas oligodendrogliomas develop by acquiring IDH mutations followed by oligodendrogliomas are caused by IDH mutation followed by deletion of 1p19q and mutation of the TERT promoter, followed by mutation of CIC.

Telomerase consists of an RNA constituent and a reverse transcriptase component. Thus, either the functioning or transcription of aberrant TERT can be a druggable target. Agents such as RNA-dependent RNA polymerase (RdRP) inhibitors [46] and FOS inhibitors that target the FOS/GABP/mutant TERT promoter cascade have been developed [47]. TERT may have functions entirely independent of telomere biology. For example, TERT is capable of performing RdRP functions via its association with BRG1, an SWI/SNF-related chromatin remodeling protein, and nucleostemin, a GTP-binding protein, forming the TERT–BRG1–NS complex (TBN complex). The TBN complex synthesizes double-stranded RNAs that are processed into short-interfering RNAs that regulate heterochromatin assembly and mitotic progression [48]. Inhibitors targeting these non-canonical functions of TERT serve as promising anticancer agents. One such inhibitor, Eribulin is known to target the RdRP activity of TERT. TERT mutations upregulate TERT expression by creating specific binding sites for the GABPA/GABPB transcription factor complex [49]. FOS, as a powerful transcriptional factor for GABPB, upregulates the expression of GABPB, which in turn binds and activates the mutant TERT promoter [50]. Targeting FOS in the FOS/GABP/mutant TERT cascade might be an effective therapeutic strategy for gliomas harboring TERT promoter mutations.

### 4.3. BRAF

BRAF, a type of oncogene that is well known as a fusion gene with UPF0606 protein-coding gene, KIAA-1549 (https://www.uniprot.org/uniprotkb/Q9HCM3/entry, accessed on 20 March 2023) or the V600E substitution, is involved in tumor formation via activation of the MAPK pathway. Generally, V600E is frequently observed in tumors, including gangliogliomas, pleomorphic xanthoastrocytomas, dysembryoplastic neuroepithelial tumors, and approximately 10% of glioblastomas. The BRAFV600E mutation results in the constant activation of intracellular BRAF molecules, which overactivates the MAPK pathway, an important signaling pathway in malignancy, and promotes tumorigenesis. Additionally, this stimulation pathway also includes bypasses through CRAF.

BRAF inhibitors have already been used in the treatment of melanoma, non-small cell lung cancer, and high-grade and low-grade gliomas in adults and children. Excellent tumor control effects for each have been shown, and recently, the combined use with mitogen-activated extracellular signal-regulated kinase (MEK) inhibitors has improved tumor control [51,52] (Figure 2). However, BRAF and MEK inhibitors still have issues with drug resistance and relapse within a few years. It has been reported that RAFi resistant tumor cells acquire resistance due to increased CRAF activity via a transition from BRAF to CRAF-dominated bypass signaling, and the use of CRAFi in combination or pan-RAFi has been used to improve therapeutic efficacy.

### 4.4. MGMT

The promoter methylation of *MGMT*, coding for a DNA repair enzyme that eliminates the alkyl group from guanine O^6^ in DNA, is considered a typical epigenomic abnormality involved in brain tumor development. MGMT promoter methylation, which has been observed in 45–75% of WHO grade 4 glioblastomas, is regarded as a positive prognostic predictor. Thus, it has been used to predict the therapeutic effect of the alkylating agent, temozolomide (TMZ), which is a standard therapeutic agent for glioblastoma. TMZ forms O^6^-methylguanine (O^6^-meG) by alkylating the DNA guanine, which induces cell death by repeating the mismatch repair cycle. However, aberrant MGMT expression removes the methyl group of O^6^-meG, and therefore diminishes the antitumor effect of TMZ (Figure 3).

Additionally, MGMT promoter methylation is correlated with G-CIMP [54]. However, its involvement in tumor development and progression remains unclear.

Aggressive tumor resection is often performed for CNS cancers; however, this alone does not guarantee a good prognosis. Additionally, most regimens are a combination of radiation therapy and chemotherapy, with the alkylating agent TMZ that is used commonly in standard chemotherapy treatments. However, as described previously, given that >50% of CNS cancer cases exhibit resistance, depending on the level of MGMT promoter methylation, many treatment strategies centered on TMZ have been attempted. One example is the RTOG 0525/EORTC 26052-22053 trial, which hypothesized that an increase in the TMZ dose would significantly improve overall survival (OS) and progression-free survival (PFS); however, no improvements were observed.

Furthermore, trials on TMZ combination therapies using IFN-β and bevacizumab (BEV), a vascular endothelial growth factor monoclonal antibody, have begun. Specifically, for BEV, the phase 3 trials AVAglio [55] and RTOG0825 [56] showed improvements in PFS, but not in OS.

Combination therapy with IFN-β has also been used to overcome treatment resistance in tumors with non-MGMT promoter methylation. The goal was to enhance the antitumor effect of TMZ by activating p53 with IFN-β, as this promotes the binding of p53 and MGMT promoters to suppress MGMT expression. A phase 2 trial, JCOG0911, was performed after a phase 1 trial (INTEGRA study) [57]. However, the data did not show any improvement in prognosis following IFN-β addition [58].

### 4.5. H3K27/H3G34

A core histone forms an octamer and binds to DNA, leading to chromatin formation. The histone tail region at the N-terminus of the histone protein regulates transcription, including structural changes in chromatin and gene expression. Generally, DNA transcriptional regulation of a regulatory region is performed via chemical modifications, including acetylation or methylation of this region. Large exome sequencing studies have also led to the identification of numerous genetic mutations in the histone tail in children with glioblastoma [59]. Several K27 and G34 mutations of H3F3A, which encode H3.3, have also been identified, and tumors that frequently exhibit these mutations are often concentrated in the median structural regions, including the brain stem, thalamus, and cerebral hemisphere. Thus, in 2021, the WHO proposed a new class of gliomas known as pediatric-type diffuse high-grade gliomas: diffuse midline glioma (H3K27-altered) and diffuse hemispheric glioma (H3.3 G34 mutant) (Table 2).

Furthermore, the H3K27M mutation induces epigenomic changes that suppress the methyl-group transfer activity of polycomb repressive complex 2, which contains an enhancer of zest homolog 2 (EZH2), the trimethylating enzyme of H3K27 (H3K27me3). However, its involvement in tumor development and progression remains unclear. An inhibitor of JMJD3, demethylase enzyme of H3K27, can successfully suppress the growth of brain stem glioma cells [60], and a previous study showed that inhibition of the reverse mechanism, EZH2, suppresses tumor progression [61]. Further studies are necessary to elucidate the underlying mechanisms.

### 4.6. NTRK-Fusion and ROS1-Fusion Genes

NTRK encodes tropomyosin receptor kinase (TRK), which binds to ligands such as nerve growth factor (NGF), brain-derived neurotrophic factor (BDNF), and neurotrophin (NT) to transmit signals into cells.

ROS1 was initially identified as a gene that shows high homology to c-ros, a proto-oncogene of the avian sarcoma virus. The Golgi-associated PDZ and coiled–coil motif-containing protein (GOPC)–ROS1 fusion gene was found in glioblastoma U118MG [62].

When NTRK or ROS1 undergoes fusion, their signaling pathways, MAPK or PI3K pathways, are constitutively activated, causing tumor hyperproliferation [63,64]. The NTRK fusion gene has been identified in pediatric gliomas, especially pontine and high-grade gliomas [65]. In 2019, the first international joint study focusing on fusion genes of infantile gliomas reported that almost all ROS1 or NTRK fusion genes were found in hemispheric primary tumors, and over 80% of these genes were found in malignant tumors [66]. In this regard, entrectinib, an inhibitior against TRKA/B/C and ROS1 and ALK, was developed as a therapeutic target for pediatric refractory gliomas. It has been reported that entrectinib showed a high rate of tumor suppression in difficult-to-treat NTRK-positive high-grade brain tumors and is expected to have therapeutic applications in the future. Furthermore, the development of second-generation inhibitors to address problems associated with resistance acquisition is underway [67].

## 5. Discussion

This article summarizes the efforts to understand tumorigenesis and the advances made in diagnosis and treatment in brain tumors, particularly gliomas. Conventional CNS cancer treatment has been changing drastically with the research being conducted. To date, it has not been clarified whether, when, and how to optimally treat individual patients to prevent the malignant transformation of their tumors, but it is not difficult to imagine that the sequential elucidation of the acquisition of genetic mutations from the origin of tumor development as described above will lead to a more specialized treatment for individual patients. Furthermore, in our previous studies, we have shown that it is possible to use mathematical models to trace the pathway of IDH-mutation-based low-grade gliomas, from development to malignant transformation [22]. With this success, we expect to be able to reciprocate the origins of tumor development and provide more effective therapeutic intervention at more appropriate times to achieve optimal timing of treatment.

Furthermore, our research team developed a nanowire device that can be used to measure microRNAs in urine and diagnose brain tumors with high probability [34]. These results have several important practical implications. The most important aspect of treating brain tumors is the detection and initiation of treatment while the tumors are still small. However, in many patients with brain tumors, the tumor size increases considerably, and once advanced, it is difficult to completely remove the tumor surgically. Especially in malignant brain tumors that require multidisciplinary treatment, the surgical removal rate of some tumors is directly related to life expectancy, and the relationship with tumor size and surrounding structures at the time of therapeutic intervention is often directly related to prognosis. Therefore, we consider it a noteworthy finding that urinalysis can detect brain tumors in the early stages.

Progress is being made in embryological studies of the central nervous system and brain tumors, with a focus on organoid technology. Further understanding of the genetic characteristics of tumors is expected to lead directly to improved treatment outcomes. As the gene-based discoveries made so far are a miniscule fraction of cancer research, it is also expected that further analysis of tumor types will lead to the development of precision medicine that focuses on unique molecular characteristics.

## Figures and Tables

**Figure 1 genes-14-01201-f001:**
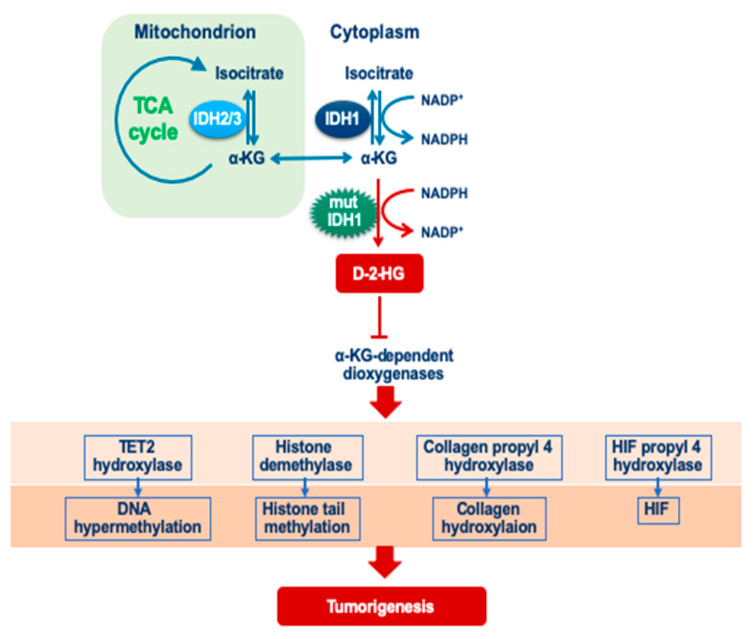
Cascades of mutant IDH1. Wild-type IDH1/2 protein uses NADP+ as a coenzyme to produce α-ketoglutaric acid (αKG) from isocitric acid, whereas mutant IDH1/2 uses NADPH as a coenzyme to produce D-2-hydroxyglutaric acid (D2HG) from αKG. Furthermore, D2HG inhibits αKG dioxygenases leading to the reduced activity of TET2 hydroxylase, histone demethylase, collagen propyl 4 hydroxylase, and HIF hydroxylase, probably as a result of the effects of IDH mutation on epigenetic regulation. All of these together contribute to tumor development.

**Figure 2 genes-14-01201-f002:**
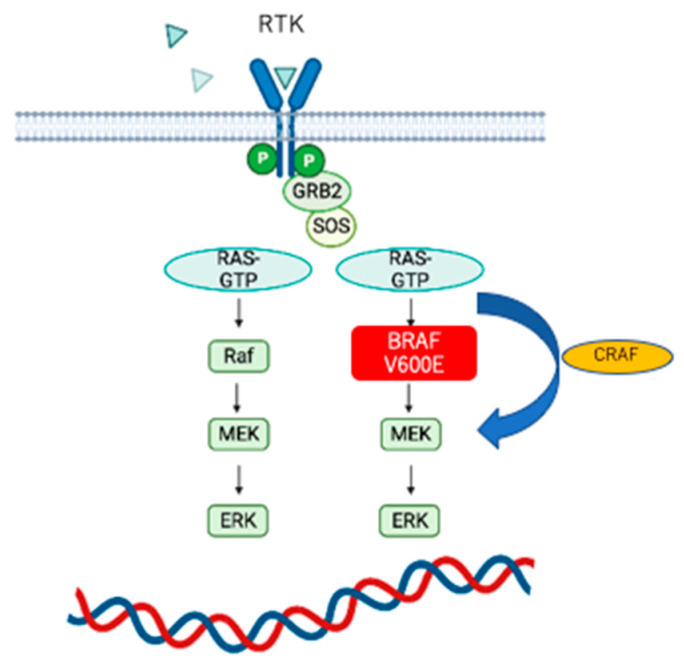
De novo cascade of BRAF V600E. BRAF V600E hyperactivates mitogen-activated extracellular signal-regulated kinase (MEK). Conversely, there is a bypass stimulation mediated by CRAF [53]. Thus, the use of BRAF inhibitors in combination with MEK inhibitors improves tumor control.

**Figure 3 genes-14-01201-f003:**
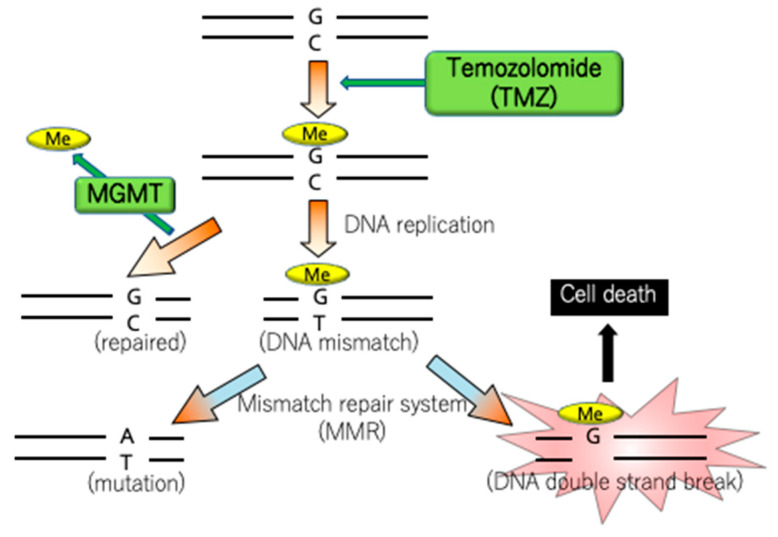
Role of temozolomide (TMZ). TMZ forms O^6^-methylguanine (O^6^-meG) by alkylating DNA guanine, which induces cell death by repeating the mismatch repair system cycle. However, in the absence of MGMT promoter methylation and when MGMT expression is not suppressed, the antitumor effect of TMZ is inhibited owing to the transfer of the methyl group of O^6^-meG by MGMT.

**Table 1 genes-14-01201-t001:** Diagnosis of adult-type diffuse glioma.

Integrated Diagnosis	Oligodendroglioma, IDH-Mutant and 1p/19q-Codeleted, WHO Grade 2 or 3	Astrocytoma, IDH-Mutant, WHO Grade 2 or 3	Astrocytoma, IDH-Mutant, WHO Grade 4	Glioblastoma, IDH Wild-Type, WHO Grade 4
IDH	Mutant	Mutant	Mutant	Wild-type
ATRX	Nuclear ATRX retained	Nuclear ATRX lost	Nuclear ATRX lost	Nuclear ATRX retained
1p/19q	Codeleted	Intact	Intact	
CDKN2A/B		Retained	Homozygously deleted **	
TERT, EGFR and/or +7/−10 *	TERT-mutant			TERT-mutant, EGFR-amplified and/or +7/−10 ***
Morphology			Necrosis and/or MVP **	Necrosis and/or MVP ***

* +7/−10, chromosome 7 gain or chromosome 10 loss. ** Integrated diagnosis is made when either two factors are present. *** Integrated diagnosis is made when either two factors are met. MVP, microvascular proliferation.

**Table 2 genes-14-01201-t002:** Diagnosis of pediatric-type diffuse glioma.

Diagnosis	Diffuse Midline Glioma, H3 K27-Altered, WHO Grade 4	Diffuse Hemispheric Glioma, H3 G34-Mutant, WHO Grade 4
IDH	Wild type	Wild type
ATRX	Nuclear ATRX retained	Nuclear ATRX > retained
H3.3 K27M-mutant (+loss of K27me3)	Mutant	
H3.3 G34R/V-mutant		Mutant

## Data Availability

No new data were created or analyzed in this study. Data sharing is not applicable to this article.

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
