# Peer review of "Prediction of Tumor Development and Urine-Based Liquid Biopsy for Molecule-Targeted Therapy of Gliomas"

_genes, 2023, doi:10.3390/genes14061201_

Round 1
Reviewer 1 Report
The draft by Kurimoto et al. provides general overview of diagnostic and theranostic advancements in CNS tumour field. Unfortunately, the work is not systematic and try to encompass too many topics. Some points are tackled in superficial way, there are also multiple inaccuracies, resulting from insufficient analysis of literature data. It is not suitable for the publication without extensive reformulation, and the careful check of the cited sources.
MAJOR POINTS:
1. The great portion of the paper lacks proper citation of the previous work. There are 30 references provided, including 7 self citations. There are whole chapters supported with one citation! For example, there is only one reference in chapter 3. It is hard to believe that only one paper worth citation devoted to the mathematical modelling of tumour development was published. It would be also very appreciated, if Authors discuss the accuracy of their own model, especially knowing that the paper was published 7 years ago.
2. The same objection can be raised for chapter 3. Authors do not cite other studies on devices/methods developed to detect CNS cancer in non-invasive way. BTW, micro RNA specific for CNS cancers was detected in patients who were already diagnosed, so no potential of wide population screening has been demonstrated yet. Moreover, what actually means that “microRNA contained in the extracellular endoplasmic reticulum in urine with nanowires… “ (line 83)? Is there any extracellular endoplasmic reticulum in urine??
3. In chapter 4, organoids derived from CNS tumour are mentioned as a tool allowing to unravel gliomagenesis. Unfortunately, Authors cite only own research, not even related to gliomagenesis! Please read Louis et al. work on CNS tumours classification (reference # 5), and acknowledge the difference between meningioma and glioma.
4. The choice of selected genes in chapter 5 is not entirely clear. Why other important genes mutated/deleted/amplified in CNS tumours, like TP53, PTEN, CDKN2A/B are not mentioned? Could Author justify the selection made?
5. Chapter 5 – some genes of interest, like EGFR, are described in very telegraphic, general way.
6. Whenever abbreviation for protein, gene name etc appears for the first time in the text, the full name should be provided.
MINOR POINTS:
7. Figure 2, lines 133 – 135. What actually means that “D2HG inhibits αKG dioxygenases leading to TET2 hydroxylase, histone demethylase, collagen propyl 4 hydroxylase, and HIF hydroxylase” Some information is obviously missing in the cited sentence. The cited source is missing.
8. Lines 144 – 146. The claim that IDH inhibitors (mutant or wt IDH inhibitors?) may be used for the treatment of wide range of cancers is quite confusing. Citations missing again.
9. Line 149. “TERT gene mutation, characterized by IDH mutation and 1p19q co-deletion” – this sentence makes no sense. The whole chapter requires reformulation.
10. Lines 162-163. The sentence is very messy. What KIAA-1549 stands for?
11. Line 176. “CRAF” need to be deciphered.
12. Subchapter on MGMT requires reformulation.
13. Lines 226-227. The information on H3 histone and histone core is inaccurate. Do these H3K27/H3G34 stand for distinct proteins?
14. Line 238. The title of Table 2 is confusing, what “IDH wild type- mutations based glioma” means?
15. Subchapter on NTRK/ROS1 fusion. There are several NTRK genes, and their products bind different neurotrophins. If the particular NTRK gene undergoes the fusion with ROS1, its name should be provided.
16. Line 274. The citation number does not match the list of references.
Author Response
Reply to Reviewer 1:
We would like to thank the Reviewer for these constructive comments. We also appreciate the suggestions, which have been immensely beneficial to our manuscript. We have also revised our manuscript in line with the Reviewer’s suggestions. Our point-to-point responses are described below.
The draft by Kurimoto et al. provides general overview of diagnostic and theranostic advancements in CNS tumour field. Unfortunately, the work is not systematic and try to encompass too many topics. Some points are tackled in superficial way, there are also multiple inaccuracies, resulting from insufficient analysis of literature data. It is not suitable for the publication without extensive reformulation, and the careful check of the cited sources.
[Our response] We agree with your comment: We have reformed our draft for more readability. For example, chapter 4 (organoid) has been omitted. We have selected the number of genes to be mentioned here from seven to six because EGFR TKIs are used for metastatic brain tumors from non-small cell lung cancers, which is not very corresponded to this review article.
MAJOR POINTS:
- The great portion of the paper lacks proper citation of the previous work. There are 30 references provided, including 7 self citations. There are whole chapters supported with one citation! For example, there is only one reference in chapter 3. It is hard to believe that only one paper worth citation devoted to the mathematical modelling of tumour development was published. It would be also very appreciated, if Authors discuss the accuracy of their own model, especially knowing that the paper was published 7 years ago.
[Our response] We have cited more recent citations, but not from ours in our revised edition. We have intensively rectified the text of Chapter 3.
- The same objection can be raised for chapter 3. Authors do not cite other studies on devices/methods developed to detect CNS cancer in non-invasive way. BTW, micro RNA specific for CNS cancers was detected in patients who were already diagnosed, so no potential of wide population screening has been demonstrated yet. Moreover, what actually means that “microRNA contained in the extracellular endoplasmic reticulum in urine with nanowires… “ (line 83)? Is there any extracellular endoplasmic reticulum in urine??
[Our response] We have cited more recent citations, but not from ours in our revised edition. We have intensively rectified the text of Chapter 3.
- In chapter 4, organoids derived from CNS tumour are mentioned as a tool allowing to unravel gliomagenesis. Unfortunately, Authors cite only own research, not even related to gliomagenesis! Please read Louis et al. work on CNS tumours classification (reference # 5), and acknowledge the difference between meningioma and glioma.
[Our response] We have omitted Chapter 4 related to organoid model, which turned out to be nonsense.
- The choice of selected genes in chapter 5 is not entirely clear. Why other important genes mutated/deleted/amplified in CNS tumours, like TP53, PTEN, CDKN2A/Bare not mentioned? Could Author justify the selection made?
[Our response] We have selected six genes to which targeted drugs are developed.
- Chapter 5 – some genes of interest, like EGFR, are described in very telegraphic, general way.
[Our response] We have omitted the paragraph mentioning EGFR.
- Whenever abbreviation for protein, gene name etc appears for the first time in the text, the full name should be provided.
[Our response] We have revised our manuscript accordingly.
MINOR POINTS:
- Figure 2, lines 133 – 135. What actually means that “D2HG inhibits αKG dioxygenases leading to TET2 hydroxylase, histone demethylase, collagen propyl 4 hydroxylase, and HIF hydroxylase” Some information is obviously missing in the cited sentence. The cited source is missing.
[Our response] We have revised our manuscript accordingly.
- Lines 144 – 146. The claim that IDHinhibitors (mutant or wt IDH inhibitors?) may be used for the treatment of wide range of cancers is quite confusing. Citations missing again.
[Our response] We have revised our manuscript accordingly.
- Line 149. “TERTgene mutation, characterized by IDH mutation and 1p19q co-deletion” – this sentence makes no sense. The whole chapter requires reformulation.
[Our response] We have revised our manuscript accordingly.
- Lines 162-163. The sentence is very messy. What KIAA-1549stands for?
[Our response] We have revised our manuscript accordingly.
- Line 176. “CRAF” need to be deciphered.
[Our response] CRAF is like BRAF, RAF c-type.
- Subchapter on MGMT requires reformulation.
[Our response] We have revised our manuscript accordingly.
- Lines 226-227. The information on H3 histone and histone core is inaccurate. Do these H3K27/H3G34 stand for distinct proteins?
[Our response] We have revised our manuscript accordingly.
- Line 238. The title of Table 2 is confusing, what “IDH wild type- mutations based glioma” means?
[Our response] We have revised our manuscript accordingly.
- Subchapter on NTRK/ROS1 fusion. There are several NTRKgenes, and their products bind different neurotrophins. If the particular NTRK gene undergoes the fusion with ROS1, its name should be provided.
[Our response] We have revised our manuscript accordingly.
- Line 274. The citation number does not match the list of references.
[Our response] We have revised our manuscript accordingly.
Reviewer 2 Report
The authors reviewed novel therapies that are being developed for central nervous system cancers. They described seven proteins, which upon mutation cause tumorigenesis and tumor progression. Their research team developed a nanowire device that can be used to measure microRNAs in urine specimens and diagnose brain tumors with high probability. It is likely that further understanding of the genetic characteristics of brain tumors will lead to the development of precise drugs and improve individual treatment outcomes.
Author Response
Reply to Reviewer 2:
We would like to thank the Reviewer for these constructive comments. We also appreciate the suggestions, which have been immensely beneficial to our manuscript. We have intensively revised our original manuscript.
Reviewer 3 Report
The review titled “Tumorigenesis and molecule-targeted therapy of gliomas” by Dr. Kurimoto et. al introduced some novel strategies and new technologies which may be helpful for the analysis of tumorigenesis, rapid and early diagnosis of tumors, and tracing tumor progression. And also summarized the roles of seven key molecules in the occurrence and progression of glioma and the development of these molecule-targeted drugs in detail. However, there are still some issues to be addressed.
Major Compulsory Revisions
1. As a review, the content of this manuscript is not sufficient. Some of the contents are more like a work summary of the previous achievements of the authors’ research team, instead of a systematical and comprehensive summary of research progress in related fields. It’s suggested that the author cite more references to enrich the content of the manuscript.
2. As mentioned in the abstract, this review mainly presents molecular-targeted therapy at first, and then it also introduces some other new technologies and strategies. But in the main body of the manuscript, the order is inconsistent with the abstract so the structure and order of the manuscript can be improved or in order from tumorigenesis to diagnosis to tumor progression estimation and to therapy.
3. Although the title focuses on “molecular-targeted therapy”, the authors did not discuss molecular-targeted therapy in depth in the discussion section, and it is suggested to further enrich it.
Minor Essential Revisions
1. According to the fifth edition of the WHO Classification of Tumors of the Central Nervous System (WHO CNS5), the old name of “Diffuse midline glioma, H3 K27M-mutant, WHO grade 4” has been modified into “Diffuse midline glioma, H3 K27-altered, WHO grade 4”, and diffuse hemispheric glioma, H3 G34-mutant is a newly added class in WHO CNS5. So the names in Table2 should be modified.
2. In Table 1, the diagnosis includes astrocytoma, oligodendroglioma and glioblastoma, so it is more appropriate to change the title to “Diagnosis of adult-type diffuse gliomas”. Similarly, the title of Table 2 should indicate that it is pediatric-type diffuse high-grade glioma.
3. In Table 1, the abbreviation of “MVP” should be expressed as a full name for the first time.
Author Response
Reply to Reviewer 3:
The review titled “Tumorigenesis and molecule-targeted therapy of gliomas” by Dr. Kurimoto et. al introduced some novel strategies and new technologies which may be helpful for the analysis of tumorigenesis, rapid and early diagnosis of tumors, and tracing tumor progression. And also summarized the roles of seven key molecules in the occurrence and progression of glioma and the development of these molecule-targeted drugs in detail. However, there are still some issues to be addressed.
[Our response]
We would like to thank the Reviewer for these constructive comments. We also appreciate the suggestions, which have been immensely beneficial to our manuscript. We have also revised our manuscript in line with the Reviewer’s suggestions. Our point-to-point responses are described below.
Major Compulsory Revisions
As a review, the content of this manuscript is not sufficient. Some of the contents are more like a work summary of the previous achievements of the authors’ research team, instead of a systematical and comprehensive summary of research progress in related fields. It’s suggested that the author cite more references to enrich the content of the manuscript.
[Our response] We have cited more recent papers, but not from ours in our revised edition. We have reformed our draft for more readability. For example, chapter 4 (organoid) has been omitted. We have reduced the number of genes from seven to six because EGFR TKIs are used for metastatic brain tumors from non-small cell lung cancers, which is not very corresponded to this review article.
As mentioned in the abstract, this review mainly presents molecular-targeted therapy at first, and then it also introduces some other new technologies and strategies. But in the main body of the manuscript, the order is inconsistent with the abstract so the structure and order of the manuscript can be improved or in order from tumorigenesis to diagnosis to tumor progression estimation and to therapy.
[Our response] The abstract has been restructured.
Although the title focuses on “molecular-targeted therapy”, the authors did not discuss molecular-targeted therapy in depth in the discussion section, and it is suggested to further enrich it.
[Our response] We agree with this comment. We have changed our title to "Prediction of tumor development and urine-based liquid biopy for molecule-targeted therapy of gliomas".
Minor Essential Revisions
According to the fifth edition of the WHO Classification of Tumors of the Central Nervous System (WHO CNS5), the old name of “Diffuse midline glioma, H3 K27M-mutant, WHO grade 4” has been modified into “Diffuse midline glioma, H3 K27-altered, WHO grade 4”, and diffuse hemispheric glioma, H3 G34-mutant is a newly added class in WHO CNS5. So the names in Table2 should be modified.
[Our response] We have revised our manuscript accordingly.
In Table 1, the diagnosis includes astrocytoma, oligodendroglioma and glioblastoma, so it is more appropriate to change the title to “Diagnosis of adult-type diffuse gliomas”. Similarly, the title of Table 2 should indicate that it is pediatric-type diffuse high-grade glioma.
[Our response] We have revised our manuscript accordingly.
In Table 1, the abbreviation of “MVP” should be expressed as a full name for the first time.
[Our response] We have revised our manuscript accordingly.
Round 2
Reviewer 1 Report
The revised version of the manuscript by Kurimoto and colleagues become more focused on one hand and gives better background for the main content of the paper on the other hand. The presented work is based largely on very own, practical experience of Authors. I suppose it explains the selection of topics and citations. Therefore, it needs to be clearly stated in the beginning that the manuscript largly presents Authors work and expresses Authors opinions regarding the future of glioma treatment.
There are also some smaller issues to resolve, mainly grammatical ones or typos, but still distracting the reader. Please find them below:
1. Line 9: In the revised version there are 6 not 7 molecules described
2. Line 16: The following sentence: “Tumorigenesis has been increasingly supported” is very unclear, what does it mean that “tumorigenesis is supported”?
3. Line 106: Past tense should be used: offers -> has offered
4. Line 181: In the following sentence: “ … amplification of which develop cancer…” change to “… , and amplification of which promote cancer development… “ is advised
5. Line 185: The choice of 6 targets should be better justified here. Why TP53, PTEN and CDKN2A/B mutation are not so interesting from the point of Authors’ view? The mutation which can be targeted (6 targets selected) are of gain-of-function type (or higher transcription in the case of TERT and MGMT), whereas TP53 etc. are loss-of-function mutation/deletion. Please acknowledge why some mutation are targetable, whereas others are not (at least directly).
6. Line 196: “… TET2 hydroxylase, stabilizes HIF… “ -> “ … TET2 hydroxylase, and in the consequence stabilizes HIF…”
7. Line 202: multiple citation of position no. 43
8. Line 207: I suggest to change “dioxygenases leading to TET2… ” to “dioxygenases, leading to the reduced activity of TET2…” – now some part of the sentence is missing
9. Line 208: “, which are probably produced as a result of the effects of IDH mutation on epigenetic regulation, which contributes to tumor development”. Please write it as a separate sentence. Just state that IDH mutations affect epigenetics and thus contribute to tumor development.
10. Line 215-216: Please specify if DS-1001 inhibits only mutated IDH1 – the information is important, if the drug targets only mutated form and doesn’t affect the healthy one.
11. Line 225: “reverse telomerase” -> “reverse telomerase subunit” (telomerase is composed of protein with RT activity and RNA – please change just for better reading).
12. Line 236: “a mutation in V600E” -> “V600E substitution”
13. Line 243: “combined use of mitogen -…” -> I would opt for: “combined use with mitogen-….”
14. FIGURE 2. The letters are hardly visible. The cell membrane here looks like the layer of cells (?). The drawing of the lipid bilayer would be more appropriate.
15. Line 253: “MGMT, the protein of which a DNA repair enzyme” -> “MGMT, coding for a DNA repair enzyme”
16. Line 263 – please use present tense (was -> is)
17. Line 263 – the information that MGMT transfers Me group to its own molecule will complete the sentence
18. Line 277: “presence or absence” -> “level”
19. Line 286: “improve” -> “overcome”
20. Line 301: “becomes” -> “forms”
21. Line 302: Removal of “It represents” and “and” from the sentence is advised.
22. Line 307: “mutations encoding” -> “mutations in”
23. Line 308: “also” is dispensable here
24. Table 2. Is it H3K27- altered or H3K27M- altered?
25. Line 319: Is this JMJD3 protein inhibitor of H3K27?
26. Line 328: Golgi
27. Line 330: “induces” -> “undergoes”
28. Line 339: “TRK/ROS1 – fusion” implies that NTRK and ROS1 actually fuse together – is it true? If not, please use “TRK or ROS1 fusions”.
29. Line 354: meningeal organoid is mentioned; however, there are also models of glioma organoids, please refer to them as well (https://pubmed.ncbi.nlm.nih.gov/34850183/, https://pubmed.ncbi.nlm.nih.gov/31883794/).
30. Moreover, citation no. 31 is probably not the right one? Organoid meningioma was described in the primary version of the manuscript and referred to the position no. 8.
Author Response
To Reviewer 1:
We would like to thank Reviewer 1 for the constructive and thoughtful comments. In accordance with your suggestions, we have revised our manuscript.
Below is the point-by-point reply to your comments:
[Major comment 1] The revised version of the manuscript by Kurimoto and colleagues become more focused on one hand and gives better background for the main content of the paper on the other hand.
[Reply] We are pleased to know that you found our revised version improved much.
[Major comment 2] The presented work is based largely on very own, practical experience of Authors. I suppose it explains the selection of topics and citations. Therefore, it needs to be clearly stated in the beginning that the manuscript largely presents Authors work and expresses Authors opinions regarding the future of glioma treatment.
[Reply] We agree with your comments. In the second revision, we clearly state that this manuscript is based mainly on our own studies and experience.
[Minor comments]
There are also some smaller issues to resolve, mainly grammatical ones or typos, but still distracting the reader. Please find them below:
- Line 9: In the revised version there are 6 not 7 molecules described
[Reply] We have corrected it accordingly.
- Line 16: The following sentence: “Tumorigenesis has been increasingly supported” is very unclear, what does it mean that “tumorigenesis is supported”?
[Reply] We have corrected it accordingly.
- Line 106: Past tense should be used: offers -> has offered
[Reply] We have corrected it accordingly.
- Line 181: In the following sentence: “ … amplification of which develop cancer…” change to “… , and amplification of which promote cancer development… “ is advised
[Reply] We have corrected it accordingly.
- Line 185: The choice of 6 targets should be better justified here. Why TP53, PTENand CDKN2A/B mutation are not so interesting from the point of Authors’ view? The mutation which can be targeted (6 targets selected) are of gain-of-function type (or higher transcription in the case of TERT and MGMT), whereas TP53 etc. are loss-of-function mutation/deletion. Please acknowledge why some mutation are targetable, whereas others are not (at least directly).
[Reply] We have corrected it accordingly.
- Line 196: “… TET2 hydroxylase, stabilizes HIF… “ -> “ … TET2 hydroxylase, and in the consequence stabilizes HIF…”
[Reply] We have corrected it accordingly.
- Line 202: multiple citation of position no. 43
[Reply] We have corrected it accordingly.
- Line 207: I suggest to change “dioxygenases leading to TET2… ” to “dioxygenases, leading to the reduced activity of TET2…” – now some part of the sentence is missing
[Reply] We have corrected it accordingly.
- Line 208: “, which are probably produced as a result of the effects of IDH mutation on epigenetic regulation, which contributes to tumor development”. Please write it as a separate sentence. Just state that IDH mutations affect epigenetics and thus contribute to tumor development.
[Reply] We have corrected it accordingly.
- Line 215-216: Please specify if DS-1001 inhibits only mutated IDH1 – the information is important, if the drug targets only mutated form and doesn’t affect the healthy one.
[Reply] We have corrected it accordingly.
- Line 225: “reverse telomerase” -> “reverse telomerase subunit” (telomerase is composed of protein with RT activity and RNA – please change just for better reading).
[Reply] We have corrected it accordingly.
- Line 236: “a mutation in V600E” -> “V600E substitution”
[Reply] We have corrected it accordingly.
- Line 243: “combined use of mitogen -…” -> I would opt for: “combined use with mitogen-….”
[Reply] We have corrected it accordingly.
- FIGURE 2. The letters are hardly visible. The cell membrane here looks like the layer of cells (?). The drawing of the lipid bilayer would be more appropriate.
[Reply] We have revised Figure 2 accordingly.
- Line 253: “MGMT, the protein of which a DNA repair enzyme” -> “MGMT, coding for a DNA repair enzyme”
[Reply] We have corrected it accordingly.
- Line 263 – please use present tense (was -> is)
[Reply] We have corrected it accordingly.
- Line 263 – the information that MGMT transfers Me group to its own molecule will complete the sentence
[Reply] We have corrected it accordingly.
- Line 277: “presence or absence” -> “level”
[Reply] We have corrected it accordingly.
From the comment 19 to 28,
[Reply] We have corrected it accordingly.
- Line 286: “improve” -> “overcome”
- Line 301: “becomes” -> “forms”
- Line 302: Removal of “It represents” and “and” from the sentence is advised.
- Line 307: “mutations encoding” -> “mutations in”
- Line 308: “also” is dispensable here
- Table 2. Is it H3K27- altered or H3K27M- altered?
- Line 319: Is this JMJD3 protein inhibitor of H3K27?
- Line 328: Golgi
- Line 330: “induces” -> “undergoes”
- Line 339: “TRK/ROS1 – fusion” implies that NTRKand ROS1 actually fuse together – is it true? If not, please use “TRK or ROS1 fusions”.
- Line 354: meningeal organoid is mentioned; however, there are also models of glioma organoids, please refer to them as well (https://pubmed.ncbi.nlm.nih.gov/34850183/, https://pubmed.ncbi.nlm.nih.gov/31883794/).
[Reply] The part about the meningeal organoid has been omitted because it is not very relevant to this review.
- Moreover, citation no. 31 is probably not the right one? Organoid meningioma was described in the primary version of the manuscript and referred to the position no. 8.
[Reply] The part of meningeal organoid has been omitted because it is not very relevant to this review.